# Kinetics of Direct Reaction of Vanadate, Chromate, and Permanganate with Graphene Nanoplatelets for Use in Water Purification

**DOI:** 10.3390/nano14020140

**Published:** 2024-01-08

**Authors:** Daniel Konradt, Detlef Schroden, Ulrich Hagemann, Markus Heidelmann, Hans-Peter Rohns, Christoph Wagner, Norbert Konradt

**Affiliations:** 1Ruhr-Universität Bochum, Fakultät für Maschinenbau und Fakultät für Chemie und Biochemie, Universitätsstraße 150, 44801 Bochum, Germany; 2Department of Waterworks, Stadtwerke Düsseldorf AG, Wiedfeld 50, 40589 Düsseldorf, Germany; dschroden@swd-ag.de (D.S.); hprohns@swd-ag.de (H.-P.R.); cwagner@swd-ag.de (C.W.); 3ICAN, NETZ Building, Carl-Benz-Straße 199, 47057 Duisburg, Germany; ulrich.hagemann@uni-due.de (U.H.); markus.heidelmann@uni-due.de (M.H.)

**Keywords:** graphene nanoplatelets, chromium(VI) removal, vanadium(V) removal, permanganate graphene reaction, graphene water dispersion, water purification, water treatment, metal-decorated graphene, heterogenous kinetics

## Abstract

Oxometalates of vanadium(V), chromium(VI), and manganese(VII) have negative impacts on water resources due to their toxicity. To remove them, the kinetics of 0.04 mM oxometalates in natural and synthetic water were studied using graphene nanoplatelets (GNP). The GNP were dispersible in water and formed aggregates >15 µm that could be easily separated. Within 30 min, the GNP were covered with ~0.4 mg/g vanadium and ~1.0 mg/g chromium as Cr(OH)_3_. The reaction of 0.04 mM permanganate with 50 mg of GNP resulted in a coverage of 10 mg/g in 5 min, while the maximum value was 300 mg/g manganese as Mn_2_O_3_/MnO. TEM showed a random metal distribution on the surfaces; no clusters or nanoparticles were detected. The rate of disappearance in aerated water followed a pseudo second-order adsorption kinetics (PSO) for V(V), a pseudo second-order reaction for Cr(VI), and a pseudo first-order reaction for Mn(VII). For Cr(VI) and Mn(VII), the rate constants were found to depend on the GNP mass. Oxygen sorption occurred with PSO kinetics as a parallel slow process upon contact of GNP with air-saturated water. For thermally regenerated GNP, the rate constant decreased for V(V) but increased for Cr(VI), while no effect was observed for Mn(VII). GNP capacity was enhanced through regeneration for V(V) and Cr(VI); no effect was observed for Mn(VII). The reactions are well-suited for use in water purification processes and the reaction products, GNP, decorated with single metal atoms, are of great interest for the construction of sensors, electronic devices, and for application in single-atom catalysis (SAC).

## 1. Introduction

The reactivity of transition metals in their highest oxidation state (oxometalates) toward carbon materials is of interest for water purification processes. Chromium(VI), Cr(VI), an inorganic water contaminant, is toxic, mutagenic, and carcinogenic to aquatic organisms and humans [1]. It is used in various industrial and commercial processes, such as chrome plating, corrosion protection, or leather tanning [2]. Due to its good solubility and low soil retention, Cr(VI) is considered a groundwater pollutant [3]. When treating drinking water, it must be eliminated to comply with drinking water limit values, e.g., WHO guidelines, with a limit value of 50 µg/L total chromium [4], or the EU Drinking Water Directive, with a limit value of 25 µg/L [5]. Previously, it was believed that Cr(VI) in natural water is unstable due to its strong oxidation potential and converts to Cr(III). However, recent studies have shown that Cr(VI) is stable in the environment at low concentrations (µg/L) and neutral pH values [6,7]. Furthermore, Cr(VI) can be formed from Cr(III) in sediments through manganese(III) catalysis [8]. Contaminated raw water requires special treatment to minimize drinking water concentrations [9]. Suitable techniques for drinking water treatment are ion exchange, reduction co-precipitation filtration (RCF), low-pressure reverse osmosis, and sorption on activated carbon [10,11,12]. The pH-c diagram shows that, in the pH range >6.5, chromate dominates, while below pH 6.5, hydrogen chromate is the predominant species. At higher concentrations and lower pH, hydrogen dichromate dimerizes to dichromate [3]. In the first row of transition metals, chromium is situated between vanadium and manganese, which also form oxometalates and have numerous technical applications [13,14]. Vanadium(V), V(V), compounds are used, for example, in catalytic and metallurgical processes. All vanadium(V) compounds are classified as toxic [15] and V(V) is a “substance of concern” according to the EPA Drinking Water Contaminant Candidate List (CCL). Its water chemistry is complex and depends on pH and concentration; Chen and Liu found the monomers [H_2_VO_4_]^−^ and [HVO_4_]^2−^ as the main species in a 0.02 mM solution in the pH range of 7 to 9 [16]. With increasing concentration, a tetramer, [V_4_O_12_]^4−^, appears at 0.2 mM, which becomes more significant with increasing concentration, which is consistent with the results of McCann and co-workers in 50 mM V(V) solution [17]. Manganese is used in many technical applications, e.g., in steel, batteries, and as a catalyst in chemical reactions. Permanganate, Mn(VII), is used primarily as an oxidizer and disinfectant. The purple permanganate, [MnO_4_]^−^, is the only species in the water and reduction occurs via the unstable manganese (VI) and (V) at neutral or alkaline pH, often to manganese dioxide [18]. Although potassium permanganate is considered an essential disinfectant by the WHO, there is evidence that lower oxidation states of manganese in the environment are problematic [19].

Since its discovery by Novolesow and Geim in 1984, graphene has received great attention [20] because it is interesting from theoretical, physical, and engineering perspectives. Graphene consists of honeycomb-fused benzene rings with above- and below-plane delocalized π electrons, resulting in a small bandgap semiconductor with a band overlap of 0.16 meV [21,22] and a carbon framework with high mechanical strength [23]. Staggered graphene layers form graphite as a semi-metallic bulk material. The band gap increases with the number of graphene layers, and with 11 layers, the difference to graphite (41 meV) is only 10%, which shows that this nanomaterial property is largely lost [24]. As a carbon material, graphene is a reducing agent, evident from reactions with oxidative reagents such as oxygen at elevated temperatures [25], ozone with photochemical activation, and hydrogen peroxide [26,27,28]. In contrast to pure oxygen, it was found that oxygen sorption from ambient air is only partially reversible [29]. Single-layer graphene is not common as a bulk material. Often, multiple graphene layers are stacked on top of each other to form few-layer graphene. If the lateral extension in two dimensions is significantly larger than the thickness, this material is referred to as nanoplate or nanoplatelet [30]. Graphene nanoplatelets (GNP) from XG Science have been patented by Drzal and Fukushima [31], are well-characterized [32,33], and are available with a sufficiently large specific surface area of 750 m^2^ g^−1^, while the maximum value of single-layer graphene is ~2630 m^2^ g^−1^. These GNP are fabricated by microwave exfoliation of acid-intercalated graphite and consist of stacks of up to ten graphene layers with a diameter of <5 µm. While the process ensures that the basal planes remain intact, the edges are substituted by hydroxyl and carboxyl groups during the process, making the material more polar than few-layer graphene [34].

Graphene oxide (GO) is the product of the oxidation of few-layer graphene, in which, in addition to edge modification, the basal carbon plane is partially oxidized [35]. It is easily dispersed in water and is often investigated for water purification, especially because the polar groups can interact with target molecules [35]. Reduced graphene oxide (rGO) is prepared by reductive or thermal treatment of GO at ~1000 °C, partially restoring the aromatic structure but exhibiting an irregular wrinkle structure compared to single- or few-layer graphene. Based on their properties, GNP, GO, and rGO should basically be suitable materials for water treatment applications [36]. Health aspects must be considered, particularly when preparing drinking water. Nanomaterials could penetrate the cell membrane and trigger cell defects [37,38]. Therefore, care must be taken to completely remove nanoparticles from the water for use as drinking water. On the other hand, it cannot be ruled out that nanomaterials will be used in drinking water treatment in the future, as shown by the example of mesoporous silica, which is used as a food additive (E551) [39,40]. There are many studies in the literature on Cr(VI) removal using carbon nanomaterials for water purification [31,32,33,34,35,36,37,38,39,40,41,42,43], but the materials have mostly been modified to improve the capacity, selectivity, and dispersibility. For example, Mondal and Chakraborty studied Cr(VI) removal using GO with an initial concentration of 5 to 80 μg L^−1^ at pH 4 to 8 and found maximum removal at pH 4. The equilibrium concentrations of Cr(VI) were evaluated with the Langmuir model and showed a maximum adsorption capacity of 1.2 mg/g [44]. The kinetics could be described as pseudo first-order adsorption (PFO). Goharshadi and Moghaddam studied Cr(VI) sorption on graphene nanosheets, but the material is better described as GO due to its high magnesium and oxygen content. In this case, the experimental equilibrium adsorption capacity was 1.66 mg/g and the kinetics was evaluated as pseudo second-order adsorption (PSO) [45]. A theoretical DFT study on graphene interaction with Cr(VI) by Hizhnyi et al. revealed that a covalent bond between chromate and graphene is energetically favored over van de Waals forces [46].

In contrast to Cr(VI), there are no data available for the sorption of V(V) with graphene, although there is interest in graphene/VO_2_ composites as battery material and electronic devices [47,48]. For example, Song et al. synthesized rGO/VO_2_ nanocomposites as anode material via a sol gel-assisted hydrothermal process by chemical reduction of V_2_O_5_ [49].

Graphene composites with Mn_2_O_3_ and MnO_2_ find application in battery and supercapacitor materials [50,51]. Zhu et al. used a redox reaction of rGO and KMnO_4_ in a pH-neutral solution under microwave irradiation to prepare graphene/MnO_2_ composites for use in supercapacitors [52]. The theoretical background was provided by Liu and co-workers, who found that hydroxyl groups are introduced into graphene through hydrolysis of permanganate ester, and the oxidative unpacking of defect-free graphene occurs simultaneously from the edge and the inner plane [53]. Afkhami and Conway studied the adsorption and electrosorption of Cr(VI), Mo(VI), W(VI), V(IV), and V(V) ions from water samples at low temperatures on large-area carbon-tissue electrodes and found an irreversible reduction, except for molybdenum [54].

We were interested in the removal of oxometalates from water resources by GNP as well as the kinetics of the process in natural water and in synthetic water as a reference [55]. Our main interest was the removal of Cr(VI), but V(V) and Mn(VII) were also considered as potential contaminants. Furthermore, it is of interest to compare neighboring oxometalates of the first period of the transition elements in their behavior towards the GNP. If possible, the reaction products should be characterized by TEM, XPS, and XRD. The redox potentials of oxometalate solutions were measured to correlate with the kinetic data. It is known that graphene absorbs oxygen and water upon contact with the surrounding air, thereby changing its properties [29], but it could be regenerated by thermal treatment [56]. Therefore, the behavior of the oxometalates towards regenerated GNP (GNP reg) was also tested.

## 2. Materials and Methods

### 2.1. Materials

All reagents, ammonium mono vanadate (1.01226), potassium chromate (1.04952), potassium permanganate (1.05082), ammonium chloride (1.01145), and sodium sulfate (1.614807), were of analytic grade and purchased from Merck KGaA, Darmstadt, Germany. Deionized water (DI) was made with a 2-stage Berkefeld system from Veolia Water Technologies Deutschland GmbH, Celle, Germany, and had a conductivity of >17.6 M Ω/cm.

The drinking water (DW) can be considered as hard and well-mineralized water with an electric conductivity of ~750 µS/cm, pH of ~7.5, and an oxygen content of 6 mg/L. The most important ion types (mean values) were calcium (82.5 mg/L), magnesium (11.5 mg/L), sodium (49.1 mg/L), chloride (104 mg/L), sulphate (54.8 mg/L), and hydrogen carbonate (198 mg/L). Due to active carbon filtration in the drinking water plant, the DW had a low total organic carbon content (<0.3 mg/L).

Synthetic water (SW) was a 5 mM solution of ammonium chloride or a 3.1 mM solution of sodium sulfate in DI, which showed an electric conductivity of ~750 µS/cm and was adjusted with 0.1 mM sodium hydroxide solution (Titripur, 1.9141, Merck KGaA, Darmstadt, Germany) or 0.1 mM hydrochloric acid (Titripur, 1.09060, Merck KGaA) to pH of ~7.5.

Graphene nanoplatelets (xGnP^®^, grade C, thickness < 4 nm, lateral size < 2 µm) with surface area of 750 m^2^/g were purchased from Sigma-Aldrich, Darmstadt, Germany, manufactured by XG Sciences, Lansing, MI, USA, and used as they were received.

### 2.2. Methods

The dynamic light scattering measurements (DLS) of the GNP were carried out using a Beckman Coulter LS Particle Size Analyzer in water with ultrasonic homogenization for 60 s before the measurement. Solid samples were analyzed with XPS using a VersaProbe II by Ulvac-Phi, which is equipped with a monochromatized Al Kα light source with 1486.6 eV photon energy and a beam diameter of 100 µm. The angle between sample and analyzer was 45°. TEM and TEM-EDS were measured with a JEOL JEM–2200 FS FE–TEM, 200 KV. Raman spectra were obtained using a Renishaw high-resolution confocal Raman micro spectrometer (excitation: 785 nm diode, 457/514 nm Ar ion and 633 nm He-Ne lasers, detection: thermoelectric (TE) cooled 1024 × 256 CCD detector with a spectral resolution of 0.5 cm^−1^ at 680 nm using the 1800 mm^−2^ grating). Infrared spectra were recorded with a Bruker Vertex 80 ATR-FTIR. The XRD pattern was obtained using a Bruker D8 Advance powder diffractometer with Cu Kα radiation (λ = 1.5418 Å) in the range of 2θ = 5–90° with a step size of 0.01° and a counting time of 0.6 s measured. To minimize background scattering, the powdered sample was spread on a Si single crystal with ethanol.

UV-vis measurements were performed using a Lambda 35 spectrophotometer, PerkinElmer Life and Analytical Sciences, Shelton, CT, USA, equipped with a 50 mm flow cell (high-performance quartz glass 200–2500 nm, No. 176,700, volume 2.25 mL, Hellma GmbH & Co. KG, Müllheim, Germany).

The elemental analysis was carried out in the Mikrolab, Oberhausen, Germany, using a CHNS analyzer (Vario EL Micro Cube from Elementar Analysensysteme GmbH, Langenselbold, Germany). Additional elements were determined after microwave digestion (MARS 6, CEM GmbH, Kamp-Lintfort, Germany) with ICP-OES (Spectro Arcos, Spectro Analytical Instruments GmbH, Kleve, Germany).

Multielement analysis in aqueous solution was performed according to ISO 17294-2 [57] using ICP-MS (Model 7900, Agilent, Santa Clara, CA, USA). The weekday calibration was carried out using certified multi-element standards (see Appendix A). Analytical properties and measurement uncertainties for ICP-MS are provided in Appendix A.

Electrical conductivity (E.C.), pH value, redox potential (ORP), dissolved oxygen, and temperature were measured and recorded with a Multi 3430 IDS connected to TetraCon^®^ 925, SenTix^®^ 980P, SensoLyt^®^, SenTix^®^ ORP 900P or FDO^®^ sensors from WTW, Weinheim, Germany.

The kinetic measurements of carbon materials were carried out on a Perkin Elmer Lambda 35 photometer with WIN-LAB software, version 6.0.3 in kinetic mode. The water sample (DW or SW) was placed in a three-necked flask equipped with sensors for temperature, pH, electrical conductivity, or optical oxygen sensor. A ceramic or metallic filter frit (5 µm) was inserted and the filtrate was continuously pumped with a cassette pump (model 205U Watson Marlow, Rommerskirchen, Germany) through a Tygon tube (SC0017T, Ismatec, Wertheim, Germany) at a flow rate of 9 mL/min through the cuvette set up in the photometer and then back into the flask. After stabilization, the background was recorded and zeroed. The UV-vis spectrometer was set to the respective ligand metal charge transfer wavelength (LMCT) in Table 1, slit width 1 nm, running time 30 min with one absorption measurement per second and the background was measured.

Analytical properties and measurement uncertainties for UV-vis spectrometry are provided in Appendix A. The calibrations can be adjusted using linear functions. The ligand metal charge transfer (LMCT) absorption bands are pH-dependent (see Appendix A).

For mass determinations in the range from 10 to 1000 mg, an XS204 analytical balance (repeatability 0.1 mg) from Sartorius, Göttingen, Germany, was used. Stock solutions of ammonium vanadate (5 mM), potassium chromate (10 mM), and potassium permanganate (10 mM) were prepared by dissolving the respective salt in DI and stored in the dark. The stock solution was added to the water (DW or SW) to obtain a 0.04 mM oxometalate solution. The run was started with stirring (800 rpm) at (25 ± 1) °C for equilibration. After 3–4 min, a certain mass of GNP was added and the time-dependent decrease in oxometalate absorption was followed by UV-vis. In salt-enriched water, nanoparticles aggregate into microparticles that can be easily filtered in-line. The decrease in oxometalate concentration was calculated as c_t_/c_0_, where c_0_ represents the initial concentration in the water before GNP addition. After the reaction was completed, the precipitate was filtered (0.2 μm), washed three times with deionized water, and air-dried. The solid was used for TEM-EDS, XPS, XRD, and desorption experiments. The filtrate was acidified to 1% *v/v* nitric acid and analyzed for residual element concentrations using ICP-MS.

In case of oxygen sorption on GNP, a 150 mL three-necked flask equipped with measurements of temperature, pH, and oxygen was filled with 140 mL oxygen-saturated water, then 300 mg GNP were dispersed, and the flask was filled completely with water. The time-dependent oxygen data were evaluated.

For desorption experiments, 460 mg GNP loaded with 0.4 mg/g vanadium, respectively, 0.9 mg/g chromium, or 45 mg GNP loaded with 10 mg/g manganese were dispersed in 10 mL DI with 150 mg sodium sulfate for 3 h. The suspension was filtered through 0.2 µm (Minisart RC, hydrophilic, Sartorius, Göttingen, Germany); the filtrate was acidified with 1% *v/v* nitric acid and analyzed for residual elemental concentrations by ICP-MS.

The oxidation/reduction potentials (ORP) of oxometalates were measured in DW with air contact. Before use, the ORP electrode was checked with ORP buffer 220 mV, pH 7 (U_H_ = 427 mV), article no. 51,350,060 (Mettler-Toledo GmbH, Greifensee, Switzerland) and redox buffer 475 mV (U_H_ = 682 mV), article no. 238,322 (Hamilton Bonaduz AG, Bonaduz, Switzerland). ORP measurements were performed by placing the sample (250 mL DW) in a three-necked flask equipped with the ORP electrode, thermometer, and pH sensor. The ORP was equilibrated in the respective water for up to 20 min with stirring (200 rpm). The measurement was then started at one measurement per minute for 15 min to record the matrix background. Then, 0.04 mM oxometalate was added to the water and the change in redox potential was measured for 15 to 30 min until the difference between successive measurements was less than 2 mV. The average value was calculated from 5 consecutive measurements and the difference was attributed to the influence of the oxometalate.

For thermal regeneration of GNP, a tube furnace (model R50/250/12, Nabertherm GmbH, Lilienthal, Germany) equipped with a quartz glass tube with adjustable permanent nitrogen flow was used. The GNP (600 mg) were weighed in a quartz crucible and placed in the unheated part of the oven and, after temperature equalization, moved to the hot zone (1050 °C) [56]. After 30 s, the crucible was pulled out and allowed to cool in the unheated zone. The GNP reg were transferred to a nitrogen-flushed Schlenk flask and stored until use.

## 3. Results

### 3.1. Characterization of Graphene Nanoplatelets

According to elemental analysis, the main elements in GNP were 89.21 mass% carbon and 8.85% oxygen by mass difference (c.f. Appendix A). The other elements were less than 1% by mass, based on dry matter. The elemental composition according to XPS was 92.1 mass% carbon and 7.9 mass% oxygen, while no other elements were detected. With respect to the small analyzed area (100 µm beam diameter), this agrees well with the results of the elemental analysis. Drzal and co-workers reported a higher carbon content of 96.7 mass% and a lower oxygen content of 3.3 mass% [61]. The obtained carbon and oxygen 1s spectra are best fitted by considering small amounts of hydroxyl [62], carbonyl [63], and carboxyl groups [64] bound to carbon beneath the strong band relating to graphene-like carbon (see Appendix A).

With TEM, GNP are semi-transparent. Fringe structures at the edges show stacks of up to 11 graphene layers for the XG graphene platelets and partially disturbed structures of the upper graphene layer (c.f. Appendix A). The Raman spectra of GNP and GNP reg showed absorption bands due to one-phonon scattering processes at 1576 cm^−1^ (G), the defect-assisted D band at 1342 cm^−1^, and the D’ band at 1619 cm^−1^. They were accompanied by the weak band of two-phonon processes at 2692 cm^−1^ (2D, G’) and 2435 cm^−1^ (D + D’’, G*) (c.f. Appendix A). The intensity-to-height ratio of I_D_/I_G_ was equal to 0.98 for GNP and 0.92 for GNP reg, which is indicative of multilayer graphene and some disorder [65,66]. Drzal et al. reported an I_D_/I_G_ of 0.12, which is characteristic of a highly ordered material [61]. Infrared spectra of GNP were recorded fresh and aged for 7 months (c.f. Appendix A). For the latter, a broad peak was found in the range of 3000–3800 cm^−1^ with a peak maximum of 3435 cm^−1^, which can be assigned to the stretching vibrations of water (∼3650 to ∼3200 cm^−1^) [67]. This peak was also measured by Drzal, but already in unaged GNP [61].

### 3.2. Dynamic Light Scattering Measurements (DLS) of GNP in Water

GNP from XG Science were dispersible in water, which is not common for all graphene materials [68]. Measurements of particle sizes and distributions of dispersions in deionized water (DI) and drinking water (DW) were carried out by DLS (Table 2). The ultrasound treatment leads to the disintegration of coarsely distributed particles and is intended to represent a measure of the minimum size of the aggregates in the corresponding dispersant.

While the GNP in DI were smaller than 1 µm, which corresponds to the specification (<2 µm), the particles in DW (E.C. 750 µS/cm) were about 50 times larger based on the mean value (>15 µm). These microparticles could be separated using a membrane or deep bed filter. A ceramic filter frit was used for the kinetic measurements.

### 3.3. Influence of Dispersion of Graphene Nanoplatelets on Water Properties

When GNP or GNP reg were brought into contact with water (DW or SW), there was a shift in pH and electrical conductivity. The effect was measured for 500 mg GNP in DW and SW as a reference for vanadium (V) and chromium (VI) kinetics and 50 mg GNP for manganese (VII). The results are shown in Table 3.

Negative pH shifts were observed for the GNP in the DW, which increased in low-buffered water, for example, in NH_4_Cl solution, or unbuffered water, as in the case of Na_2_SO_4_ solution. The dispersion of GNP in DW resulted in a reduction of E.C. that was not observed for SW. In contrast, the use of GNP reg resulted in a positive pH shift, probably through sorption of the carbon dioxide present in DW and SW, but did not result in a significant increase in E.C. These phenomena must be considered during the kinetic evaluation and the evaluation time can only begin after complete dispersion.

### 3.4. Sorption of Oxygen by Graphene Nanoplatelets

Graphene is known to adsorb oxygen and water upon contact with ambient air [29]. This also applies to the GNP used. A maximum weight gain of 0.34% was measured in the first hour, which increased to 2.67% after 24 h (c.f. Appendix A). In order to minimize the aging process, it is therefore important to exclude oxygen and water from the solid GNP during storage as far as possible. Regenerated GNP were stored under nitrogen.

When GNP were dispersed in oxygenated water, the oxygen was partially sorbed, a process that could be monitored with an oxygen sensor. Kinetic data for the sorption were collected for GNP and GNP reg in DW and SW; oxygen concentration–time profiles with empirical function, pH, and EC curves are documented in Appendix A. The data obtained were evaluated in terms of adjusted R^2^ (R^2^ adj), maximum evaluation time, and best fit according to PSO kinetics (Table 4). Data used, fitted parameters, and fitting quality indicators are provided in Appendix A. The oxygen sorption is also present during the sorption of oxometalates and the relevance will be discussed in Section 4.

### 3.5. Sorption of Vanadium(V), Chromium(VI), and Manganese(VII) on GNP

GNP reacted in water at 25 °C with V(V), Cr(VI), and Mn(VII). The reaction progress can be monitored by UV-vis spectrometry (Figure 1) on the specific LMCT absorption wavelengths (Table 1). The attributes of the kinetic runs are summarized in Table 5. Concentration–time profiles with an empirical function for curve fitting as well as pH and E.C. curves are provided in Appendix A.

The reaction of 0.04 mM permanganate with 50 mg (4.16 mmol) GNP was complete after 5 min, while the sorption for 0.04 mM vanadate or chromate on 500 mg (41.6 mmol) GNP was largely completed after 30 min. The final metal concentrations in the filtrates measured by ICP-MS analysis confirmed the decrease in the corresponding metal in the suspension (c.f. Appendix A). After fixing the metal on the GNP, capacities of ~0.4 mg/g for vanadium (with GNP reg ~0.9 mg/g) and ~1.0 mg/g for chromium (with GNP reg ~2.0 mg/g) were obtained. For permanganate, a capacity of ~10 mg/g manganese was measured, while the total value could be determined after multiple Mn(VII) dosages of up to 300 mg/g (Table 5).

To standardize the influence of cations and anions in drinking water, the sorption of oxometalates was also monitored in synthetic water (SW); for V(V) and Cr(VI), 5 mM ammonium chloride solution was used, and in the case of Mn(VII), a 3.1 mM sodium sulfate solution, as ammonium chloride is partially oxidized to nitrate by permanganate.

Due to aging, thermal regeneration was expected to improve the reactivity of GNP toward oxometalates. Therefore, GNP regenerated at 1050 °C were tested for the sorption of oxometalates with DW and SW.

The sorption of oxometalates on GNP in DW and SW was accompanied by negative pH shifts, especially in the low-buffered SW (Table 5). Comparable pH shifts also occurred when GNP were dispersed alone in the dispersant (c.f. Table 3). The pH value had an influence on the intensities of the UV-vis absorptions (c.f. Appendix A), but for the evaluation range of kinetics (c.f. Appendix A), there were only small errors in the concentrations of V(V), Cr(VI), and Mn(VII). Even the large ∆pH change of +2.21 of Mn(VI) with GNP reg in 3.1 mM Na_2_SO_4_ solution led in this case to an error of <0.1% due to the very low pH dependence of the absorption band at 525 nm. Elemental analysis data of the metal-decorated GNP are provided in Appendix A.

### 3.6. Evaluation of Kinetic Schemes

For the interpretation of experimental data from sorption processes, two main models are discussed, that is, pseudo first-order (PFO) and pseudo second-order sorption (PSO), the importance of which has been re-evaluated by Ho and McKay [69]. The difference between PFO/PSO and pseudo-first- and pseudo second-order reactions was highlighted by Tran [70]. In particular, the PFO model can be used to describe experimental data where the underlying process is physisorption. It is based on electrostatic interactions between the adsorbent and the adsorbate, primarily van der Waals forces. In contrast, the PSO model describes experimental data well when the underlying mechanism is chemisorption where only a monomolecular layer can be absorbed and sorbents such as sorbate change their binding structures.

For evaluation, the UV-vis kinetic data of oxometalates with GNP were measured at a frequency of one per second, reduced to a maximum of 40 data pairs for regression analysis and adapted to different models, e.g., first-order reaction and pseudo second-order reaction, PFO and PSO [71]. The linearized formulas according to Table 6 were used for the evaluation. The linear parameter fit was based on the routine according to Bevington and Robinson [72].

As model performance indicators, the adjusted coefficient of variation (R^2^ adj), which takes into account the number of measurements, was calculated, and a minimum value of 0.9 was assumed. In addition, a graphical analysis of the residuals [71] was carried out, paying attention to an even distribution of the residuals in terms of size and sign (non-randomness and normality test). Under these conditions, only one model for each oxometalate was reasonable. Regression data can be found in Appendix A. The resulting rate constants are summarized in Table 7.

After an initial phase in which the GNP are dispersed, vanadium(V) sorption can be described with PSO adsorption kinetics over a long period of time. The sorption of Cr(VI) with GNP is best described using pseudo second-order reaction kinetics. In case of Mn(VII) sorption with GNP, the concentration–time profiles can be fitted with high R^2^ adj values to a pseudo first-order reaction for the rate-determining step. For V(V) sorption in SW, the rate constant was slightly increased (factor 1.4) compared to DW, while the capacity approximately doubled. The Cr(VI) reaction with GNP was 16 times faster in SW than in DW with comparable capacity (factor 1.1). For Mn(VII) reaction with GNP in SW, the rate constant was reduced to 64% (reference DW), which is due to the high sulfate concentration in SW.

For the regenerated GNP, the rate constant for V(V) in DW was reduced by a factor of 5.5, while the capacity in DW was increased by a factor of 1.8; in SW, the rate constant was reduced by a factor of 6.2 and the capacity was reduced to 94%. With Cr(VI) in DW, the rate constant was greatly increased (factor 65) and the capacity was also increased by a factor of 2.3; in SW, the rate constant was increased by a factor of 4.5 and the capacity was doubled. For Mn(VII) in DW, no influence of regeneration on the rate constant was found, while in SW, the rate constant was increased by a factor of 1.5. The maximum capacities of manganese on GNP reg have not been determined.

### 3.7. Graphene Nanoplatelet Mass Influence on Oxometalate Sorption

It is of interest how the mass of the GNP influences the reaction rate at constant oxometalate concentration. For 0.04 mM V(V) in DW, four masses between 500 and 1000 mg were applied; for 0.04 mM Cr(VI) in DW, the mass of the GNP was varied in six steps between 250 and 875 mg. In the case of Mn(VII), five masses between 10 and 50 mg of GNP were used. Concentration–time profiles with an empirical curve-fitting function, pH, and EC curves are given in Appendix A. The regression data with error estimation are documented in Appendix A. The summarized kinetic data are presented in Appendix A. To further evaluate the data, the obtained rate constants were plotted against the GNP surfaces, which are calculated with the GNP specific surface area of 750 m^2^/g.

For vanadium(V) sorption, plotting the logarithm of the PSO rate constant against the surface of the GNP leads to a regression line with a small, negative slope (Figure 2). The *p*-value was calculated using the statistics program GRETL [74,75] to be 0.019 and is above the significance level of α = 0.01, which shows that the linear fit on the high level is not secured. The tendency of log k_PSO_ to decrease with increasing surface area could be due to the formation of larger particles with reduced surface area with increasing GNP concentration.

In contrast, for chromium(VI), a regression line with a positive slope is obtained when the logarithm of the pseudo second-order rate constant is plotted against the surface of the GNP (Figure 3). The *p*-value for the slope was calculated to be 1.72 × 10^−6^ [75], which is well below the significance level α = 0.01, showing that the linear fit is significant. The linear dependence of the surface implies that the reaction with Cr(VI) only takes place on the surface.

In the case of manganese(VII), only the double logarithmic plot of the pseudo first-order rate constant and surface area gives a linear regression line (Figure 4). This behavior is called the power law and is found in the analysis of many natural phenomena, but is susceptible to incorrect attributions. Therefore, statistical tests should be applied to the data [76]. In this case, the *p*-value for the linear fit is 2.83 × 10^−6^ and therefore well below the significance level α = 0.01, and the linear fit is significant (R^2^ adj = 0.9986). The exponential increase in both the surface area and the rate constant implies that multiple graphene layers are involved in the reaction with manganese(VII).

### 3.8. Competitive Sorption of Vanadate and Chromate

Graphene nanosheets offer different sorption sites: the electron-rich surface, layer boundaries, different types of vacancies, and the edges. Competitive sorption experiments can provide information about whether the same sorption site is preferred. The parallel sorption of 0.04 mM V(V) and 0.04 mM Cr(VI) with 500 mg GNP resulted in the time-dependent concentration curves in Figure 5. The V(V) absorption at 263 nm was disturbed by a Cr(VI) absorption, so a correction was applied (Appendix A).

At t_0_ and 1 min after GNP addition, both concentrations were the same, but at 1.75 min, the sorption of V(V) almost stopped, while the Cr(VI) followed a second-order decrease. The rate constant was (5.54 ± 0.15) mM^−1^ min^−1^ (for regression data, see Appendix A). The stop of V(V) sorption is evidence that V(V) and Cr(VI) compete for the same sorption sites, but Cr(VI) prevails against V(V).

### 3.9. Desorption of Metal Species from GNP

Particularly in water treatment, it is of interest whether the metal species are fixed on the graphene or could be desorbed under other conditions. In this case, a 15 g/L sodium sulfate solution was used according to Section 2.2 with a high concentration of the divalent sulfate anion, which should be able to desorb monovalent and divalent anions. In the filtrate of the desorption experiments of occupied GNP (0.4 mg/g vanadium, 0.9 mg/g and 11 mg/g manganese), the respective concentrations were 0.38 mg/L vanadium, 0.16 mg/L chromium, and 8.6 mg/L manganese. This results in desorption rates of 2.1% vanadium, 0.39% chromium, and 17% manganese. It turns out that about 80% or more of vanadium and chromium were anchored to graphene, which is the case when a reaction product is insoluble, such as chromium (III) hydroxide. The high desorption rate of manganese is an indication that the products are partially soluble in the sodium sulfate solution. In the case of vanadium, sorption forms an insoluble vanadium compound.

### 3.10. Characterization of Metal Species on GNP

To obtain information about products of GNP with oxometalates, the decorated GNP were analyzed by XPS and TEM. Both methods can provide information about element concentrations of first-row transition elements when higher than 0.1 atom% or 0.5 mass% [77,78]. In the case of GNP with vanadium (V-GNP), the mass concentration of vanadium was below the LOQs of XPS and TEM-EDS. For the reaction product of Cr(VI) with GNP (Cr-GNP), the XPS results are shown in Table 8.

To obtain information about the bonding structure of the chromium species, high-resolution XP spectra were recorded for chromium, carbon, and oxygen. The Cr 2p spectrum in Figure 6 shows the doublet Cr 2p_1_ and Cr 2p_3_ [79]. Both the peak position and the peak shape of Cr 2p_3_ agree well with Cr(III) hydroxide; no higher oxidation states were found [80], which is evidence for the reduction of the Cr(VI) to Cr(III). The Cr 2p spectrum is similar to that obtained by An and co-workers for the reduction of chromium(VI) on activated carbon, but with the difference that they additionally identified a small amount of adsorbed Cr(VI) [81].

The XPS signals for carbon 1s and oxygen 1s are shown in Figure 7. The carbon spectrum is dominated by graphene that has been mildly oxidized, possibly at the edges [82,83]. The different carbon–oxygen species can be assigned to hydroxyl, carbonyl. and carboxyl groups, which can also be found in the corresponding O 1s spectrum. The contribution of the oxygen–chromium species is small due to the low chromium content in the sample [84].

Graphene and GNP are semitransparent to TEM, as shown in Figure 8. In the illustrated case of Cr-GNP, the left edge of the GNP flake shows four graphene layers. The EDS spectrum demonstrates the random distribution of chromium; no clusters or nanoparticles were found.

The reaction of GNP with potassium permanganate (Mn-GNP) resulted in a final product with a high manganese content of ~30% by mass. The results of the XPS analysis are shown in Table 9.

The main components are carbon, manganese, and oxygen. Calcium and chloride may result from the DW matrix (c.f. Section 2.1).

The XPS envelope in the 2p manganese region is best fitted with a mixture of Mn(II) oxide and Mn(III) oxide/hydroxide [80], with about 80% of the signal arising from Mn(III) (Figure 9). As in the case of Cr-GNP, the carbon spectrum is dominated by graphene-like carbon (Figure 10). Approximately 25% of the detected carbon signal is made up from either oxidized carbon species or some non-sp^2^-hybridized C-C/C-H bonds [83]. The oxygen 1 s peak shows strong bands at around 529.8 eV and 531.2 eV, corresponding to oxygen in Mn(II) or Mn(III) [85]. The peak at around 532.7 eV and partially the one at 531.7 eV consist of oxygen in different carbon–oxygen species. This interpretation of the binding energies also fits the calculated intensities of the different elemental species, with roughly half of the total oxygen signal arising from the manganese oxides [84] and the other half from the oxidized carbon species.

Mn-GNP were characterized by XRD (see Appendix A), but except graphitic carbon at 2θ = 26.7°, no other crystalline phase could be identified. The peak at 36.5° could correspond to a manganese(II) hydroxide, but the assignment based on a single peak is not reliable. It can therefore be assumed that the manganese compounds (Mn_2_O_3_ and MnO) are amorphous.

TEM-EDS of the GNP reaction product with Mn(VII) showed a random distribution of manganese (Figure 11). The carbon is almost completely covered by manganese.

### 3.11. Redox Potential Measurement of Oxometalate Solutions

In redox reactions, it is important to know the magnitude of the driving force, i.e., the Gibbs free energy, which is proportional to the redox potential differences between the oxometalates and the GNP (Equation (5)). Since the GNP represent a heterogeneous phase and Equation (5) can only be applied in a homogeneous phase, in this case, the relative potential change (E_ox_) caused by the oxometalates in water was determined.
∆G^0^ = −n × F × (E_red_ − E_ox_)(5)

The influence of 0.04 mM oxometalates on the redox potential (ORP) in oxygen-saturated DW was measured (see Appendix A) and the ORP shift by oxometalate dosage was calculated as the difference between the mean before and after addition (Table 10).

The ORP increased in the order V(V) < Cr(VI) << Mn(VII), with the relation of 1:1.5:11.

## 4. Discussion

At the beginning of the investigation, it was surprising that the graphene nanoplatelets could be easily dispersed in synthetic and natural water, although the dispersion in detergent-free water was described by Wang and coworkers [85]. This may not apply to all graphene nanomaterials, but it does apply to GNP by XG Science. In contrast, multilayer carbon nanoparticles obtained by spray flame synthesis float on the water phase, like soot, and cannot be dispersed even when energy, e.g., by ultrasound, is applied [86]. The dispersibility of the GNP from XG Science can be explained by increasing hydrophilicity with increasing number of graphene layers [82] and polar groups at the edges [31]. DLS showed that the GNP formed larger aggregates, for example, in deionized water <1 µm and in natural water >15 µm (mean values). In DW, the aggregates settled without stirring and could be completely separated from the water phase using a 10 μm filter, which is important because separation is necessary after using GNP in water treatment. The wetting of graphene by water, which is important for dispersibility, is complex [87]. Results of wettability of graphene with water provided a variety of results and showed that the underlying substrate has a strong influence. For example, Checco and co-workers measured a contact angle of 85° ± 5° for fully suspended graphene, compared to 61° ± 5° for partially suspended or supported graphene [88]. The change in contact angle due to storage under ambient conditions has also been described. The adsorption of water, oxygen, and air pollutants, for example, hydrocarbons, leads to an increase in hydrophilicity and the contact angle and thus to an improvement in dispersibility in water. The aging of GNP under ambient conditions is also visible in the IR spectrum (see Section 3.1), with an increase in the OH stretching vibration, which is an indication of adsorbed water. The broad IR band of the O-H stretching vibration can be interpreted as an indication of different environments of the water molecules [67]. The XPS of fresh GNP showed the presence of hydroxyl, keto, and carboxyl groups. This is consistent with data from Drzal et al. on the structure of GNP with intact basal planes with carboxyl, hydroxyl, lactone, and pyrone structures at the edges, which are partly responsible for the dispersibility in water [31]. Yang and Murali found that when graphene is doped with oxygen under ambient conditions, it is attacked from the edges or places nearby. In contrast to dry air, the process is only partial reversible, and the process increases with contact time [29]. The adsorption energy of oxygen was estimated by Kawayama et al. with temperature dependence THz emission microscopy by thermal desorption to ~150 meV (14.5 kJ/mol), which is in the range of an adsorption [89]. According to Kong et al., graphene is always p-doped due to the adsorption of water and oxygen in ambient air, but the effect decreases with the number of graphene layers [90]. Smith, Kay, and May measured the desorption energy for oxygen under UHV conditions to be about 12 kJ/mol [91]. Therefore, it is important to protect the GNP from oxygen, water, and air pollutants as much as possible. The results of this study showed that GNP in contact with ambient air led to a weight increase of 0.34% in the first hour and the process was not complete even after 24 h. An observed reduction in reaction rates, especially in the reaction with chromium(VI), is attributed to aging, which is why, for example, the dependence of the rate constant on the GNP surface was carried out close in time.

A reaction of oxygen with the GNP also takes place in water (see Section 3.4), although this occurs significantly more slowly than oxometalate sorption. With the exception of the adsorption of V(V) on GNP reg in DW and SW, all rate constant ratios are 10 times higher, even if the oxygen concentrations of ~0.25 mM were a factor of 6.3 higher than the oxometalate concentrations (Table 11). As shown in Table 4, oxygen sorption is best adapted as a PSO process (Figure 12). However, it is not certain whether the same sorption sites are occupied as for V(V) and Cr(VI).

Oxometalates of vanadium, chromium, and manganese react with GNP in water to form solid, insoluble products that are the result of redox reactions. The low oxidation potential of 0.04 mM vanadate already leads to a vanadium capacity of 0.42–0.87 mg/g. The measured potential of vanadate of +44.6 mV for a 0.04 mM solution can be compared to data obtained from cyclic voltammetry (CV). Çakir and Biçer measured the CV of a 0.05 mM NH_4_VO_3_ in phosphate buffer (pH 7.4) and assigned the peak at +110 mV to the reduction of V(V) to V(IV) [58]. Using the method described by Demsey et al. [92], an E_1/2_ ≈ E_0_ of +100 mV could be derived. This is supported by Chen and Liu, who observed a one-electron reduction to V(IV) in a 0.02 mM solution in the pH range of 7 to 9, where monomeric [H_2_VO_4_]^−^ and [HVO_4_]^2−^ species dominate [16]. The kinetics of V(V) with GNP follows the PSO kinetics and the process is thus determined by thermodynamics: diffusion through the liquid phase to the solid, adsorption, and reaction, with the diffusion process determining the kinetics. The reaction product is fixed to the surface probably as vanadium(IV) oxide (Figure 12). Unfortunately, the vanadium species bound to the GNP could not be determined, but the small amount of water-soluble vanadium in the desorption experiment of V-GNP and the one-electron reduction at low oxidation potential [58] suggest vanadium dioxide. Parallel sorption of Cr(VI) and V(V) revealed that both oxometalates compete for the same sorption sites, with Cr(VI) displacing V(V), implying a time-delayed reaction of the adsorbed vanadate with the GNP.

Heterogeneous kinetics are complex and consist of at least three steps. The first step is the diffusion of oxometalate from the bulk liquid to the GNP surface through the concentration boundary layer, sorption to the surface, followed by reaction. To describe the reactions of the oxometalates with the GNP, the shrinking surface model known from chemical reaction engineering can be used, since the surface of the GNP is reduced by being covered with product [93]. For a given surface, the mass transfer coefficient in the concentration boundary layer during adsorption or the rate constant during a reaction and the concentration of the reactant in the liquid main phase are crucial for the time-dependent mass conversion. Important factors that accelerate kinetics are:increase in the surface area of the solid, here of GNP;decreasing size of the particles, here nanoplatelets;increasing the relative liquid/solid velocity, here stirring at 800 rpm.

To compare the sorption of GNP with oxometalates, the conditions were kept constant except for manganese(VII), where mass had to be reduced to 50 mg GNP and the SW was a sodium sulfate solution.

A comparison of rate constants of V(V) between the synthetic (SW) and drinking water (DW) showed that a reaction in SW is faster by a factor 1.4 and the capacity is increased (factor 2.2). This can be explained by competition of the vanadate sorption by sulfate present in DW. With regenerated GNP, the rate constant of V(V) sorption decreased by a factor of approximately 5.5 (DW) or 6.2 (SW), while the capacity increased in DW (factor 1.8) or remained almost constant in SW. During regeneration, the water, nitrogen and oxygen adsorbed on the GNP were completely removed, and after suspension, the interfacial water layer on the GNP must be re-formed before the sorption process can begin. The interfacial water structure on graphite was studied by Garcia et al. using AFM and indicates the existence of two separate hydrophobic water layers. The distance between the graphite surface and the first layer was 0.44 nm, and between the two water layers it was 0.55 nm, and there was evidence of fluctuating structures in the bulk water [94]. Similar results were previously obtained through molecular dynamics simulations of water molecules on the surface of a single graphene layer at room temperature [95]. The surface of the double-layer structure acts as a hydrophobic surface because the hydrogen bonds are only formed within the respective layer. It should also be considered that in an aqueous environment, the oxometalates of the d-block transition metals are hydrated with two molecules of water, so that the adsorption process could begin with the formation of hydrogen bonds [96]. In addition, unpublished results show that the delay in the reaction with GNP reg is largely due to the adsorbed nitrogen, which must first be exchanged for oxygen in order for the sorption of vanadate to occur within 30 min. The influence of the GNP mass on the PSO rate constant of V(V) sorption was neglectable. A slight decrease in log k_PSO_ with increasing surface area could be explained by the formation of larger particles with reduced surface area (see Section 3.2).

For Cr(VI) in the concentration range of 0.04 mM and pH > 6.5, the chromate dominates [10]. The redox potential of 0.04 mM Cr(VI) was measured to be +67 mV, which is only 23.9 mV higher than for V(V) and resulted in a chromium capacity of 0.93–2.1 mg/g, which is comparable with the adsorption capacity of 1.66 mg/g measured by Goharshadi and Moghaddam [45]. If the ORP is the only factor, the change in kinetics occurs within this small potential difference. Since sorption followed pseudo second-order reaction kinetics, the rate-determining step is a chemical reaction. This result is in contrast to the kinetics studied by Goharshadi and Moghaddam, who prepared rGO by reducing solid carbon dioxide with magnesium and found PSO kinetics with Cr(VI) [45]. XPS identified chromium(III)hydroxide as the sole product (c.f. Figure 12). For Cr(VI), the change from DW to SW resulted in an increase in the rate constant by a factor of 16, while the capacity was only slightly increased (factor 1.1). With regenerated GNP, Cr(VI) showed a strong increase in rate constant by a factor of 65 for DW and 4.5 for SW with an increase in capacity by a factor of 2.3 (DW) and 2.0 (SW). In this case, regeneration increased the rate constant and the capacity by providing additional reactive sites by eliminating water, nitrogen, oxygen, and pyrolysis products such as carbon monoxide and carbon dioxide. In an ab initio study, Hizhnyi et al. proposed a chromium ester as a reaction intermediate in which an oxygen is bound to a honeycomb carbon atom that undergoes sp^3^ hybridization [46]. But there was a second minimum with three oxygen atoms of chromate equidistant from the graphene plane and the chromium centered over a honeycomb carbon that could be interpreted as adsorbed chromate. The first intermediate should be responsible for the observed reaction. The pseudo second-order kinetics showed that the rate constant is proportional to the square of Cr(VI) concentration, meaning that two chromate ions are involved in the rate-determining step. This can be explained by a parallel attack of two neighboring carbon atoms of graphene, which are electronically coupled via the conduction band, or by a very low equilibrium bichromate concentration in the reaction medium, which is unlikely. The linear increase in log k_2_ with the GNP surface area can be interpreted as an indication of a surface reaction. Although a pseudo second-order kinetic evaluation can be applied to chromate sorption, it is important to note that the mass of suspended GNP has an influence on the rate constant. TEM showed that chromium was randomly distributed on the GNP.

For Mn(VII), only permanganate is the relevant species in water and the measured ORP difference of ~500 mV for a 0.04 mM solution was about 7-fold higher than in the case of Cr(VI). The high oxidation potential leads to high manganese capacity of 300 mg/g. Due to the high capacity of the GNP, the reaction followed pseudo first-order kinetics at only one-tenth of the GNP mass. In contrast to Cr(VI), the reaction rate in SW was reduced by a factor of 1.6, which may be due to the sulfate content in SW (3.1 mM Na_2_SO_4_), which is higher than in DW (0.59 mM). Before the reaction, the divalent sulfate must be replaced by the monovalent permanganate, which reduces the velocity. The use of the GNP led to an increase in the rate constant in SW by a factor of 1.5 and showed no effect in DW. The double logarithmic plot of the first-order rate constant against the GNP surface was linear, which is taken as an indication of the involvement of multiple graphene layers. In this case, the electronic graphene structure should be partially destroyed. DFT calculations by Liu et al. showed a 5-ring intermediate before hydrolytic cleavage [53], consistent with pseudo first-order kinetics. They proposed the oxidation of neighboring carbon atoms as the first step to detach graphene layers from both edges and the inner plane. According to TEM-EDS, manganese atoms were randomly distributed on the GNP. According XPS, the manganese (VII) was reduced to 80% Mn(III) oxide/hydroxide and 20% Mn(II)oxide/hydroxide. The formation of Mn_2_O_3_ nanorods under hydrothermal conditions was reported by Khisro, Chen, and coworkers [97]. However, the manganese oxides/hydroxides identified in this study were amorphous according to XRD analysis.

## 5. Summary and Outlook

In order for graphene nanoplatelets to be used as a water treatment agent, they must fulfill the treatment purpose, be available in consistent quality, have an acceptable price, and be safe to handle. For most continuous water purification processes, the exposure time must not exceed 30 min and the conditioning agent must be completely removed, conditions that are largely achieved with the used GNP. With an average particle size of 15 µm in water with an E.C. of ~750 µS/m, the particles can be separated using membranes or deep bed filters.

With a surface area of 750 m^2^/g, the graphene nanoplatelets achieved capacities of 1.0 mg/g vanadium, 2.0 mg/g for chromium, and ~300 mg/g for manganese. The sorption capacities increased with the redox potential of oxometalate in the order V(V) < Cr(VI) << Mn(VII). A comparison of the rate constants in SW and DW showed that competing anions (e.g., sulfate) occupy the sorption sites on the GNP and reduce the rate constants. The thermal regeneration of GNP increased the capacities for vanadium, but reduced the rate constants, while for chromium, the capacities and the rate constants were significantly increased through the provision of additional reactive sites. For manganese, no regeneration effect was observed in DW, while in SW, the rate constant increased by a factor of 1.5.

In aerated water, vanadate sorption on GNP occurred with PSO adsorption kinetics, while chromate reacted with GNP with pseudo second-order kinetics and permanganate reacted with pseudo first-order kinetics. Thus, the three neighboring transition metal oxometalates showed different behavior. In vanadium(V) sorption, the thermodynamically controlled adsorption step was rate-limiting, followed by a surface reaction that resulted in an insoluble, non-desorbable product, probably vanadium(IV) oxide. Chromium (VI) was reduced to Cr(III) hydroxide through a kinetically controlled reaction and fixed to the graphene surface. Manganese(VII) was reduced through a kinetically controlled reaction to 80% Mn(III) oxide/hydroxide and 20% Mn(II)oxide/hydroxide, which were largely insoluble. Accordingly, vanadium, chromium, and manganese were fixed on the graphene surface through a chemical reduction process, even if an adsorption step took place beforehand. Using TEM-EDS, the distribution of chromium and manganese on the GNP was found to be random and no clusters were observed.

The sorption of oxometalates with GNP can be generalized. Initial experiments showed that GNP can also be used to remove molybdenum(VI), tungsten(VI), and uranium(VI) from the aqueous phase. For material synthesis, sorption can be controlled by determining the concentration of reactants and the reaction time to produce a product with a defined single-atom metal coverage. Unless an additional reducing reagent is used, the maximum capacity is limited by the redox potential of the oxometalate. The reaction products are GNP covered with metal compounds in a lower oxidation state compared to the corresponding oxometalates. The metal compounds can be further modified through chemical reactions such as reduction [98] or ligand exchange. The GNP skeleton can be modified by edge functionalization, basal plane functionalization, or noncovalent adsorption on the basal plane [99]. Metal-coated graphene is of great interest for the construction of sensors [100], electronic devices [101], and for application in single-atom catalysis (SAC) [102,103]. The reaction type can also be extended to hydrophobic graphene, for example, a multilayer plasma-synthesized graphene [86] that is dispersible in a binary solvent mixture of methanol and water.

## Figures and Tables

**Figure 1 nanomaterials-14-00140-f001:**
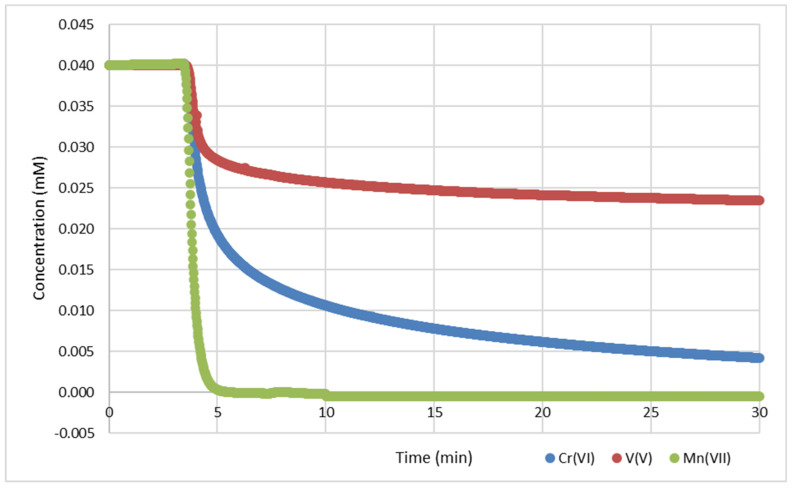
Kinetics of 0.04 mM oxometalates in drinking water (DW) with 500 mg GNP (50 mg for Mn(VII)), with UV-Vis at the wavelengths listed in Table 1.

**Figure 2 nanomaterials-14-00140-f002:**
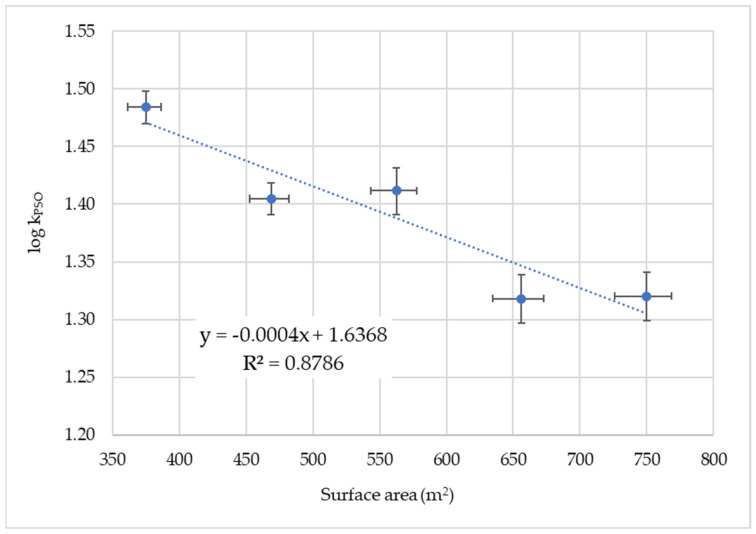
Plot of log k_PSO_ of V(V) sorption versus surface area of GNP. The error bars are the mean standard deviations.

**Figure 3 nanomaterials-14-00140-f003:**
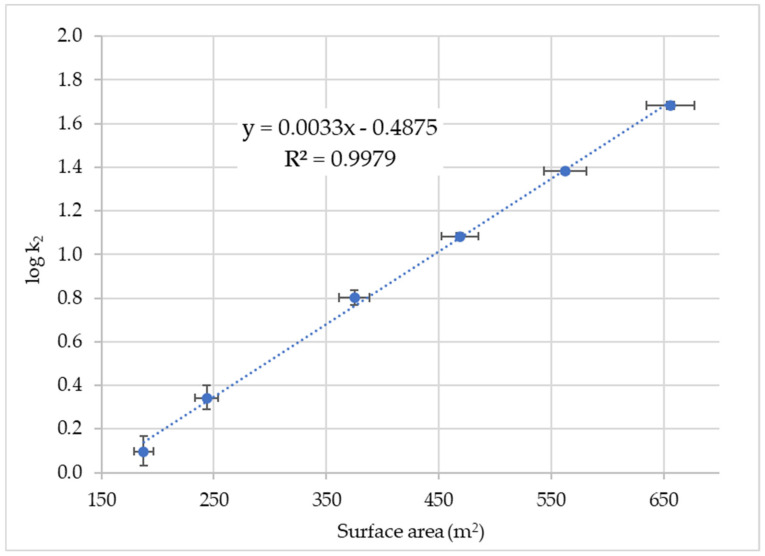
Plot of log k_2_ of Cr(VI) sorption versus surface area of GNP. The error bars are the mean standard deviations.

**Figure 4 nanomaterials-14-00140-f004:**
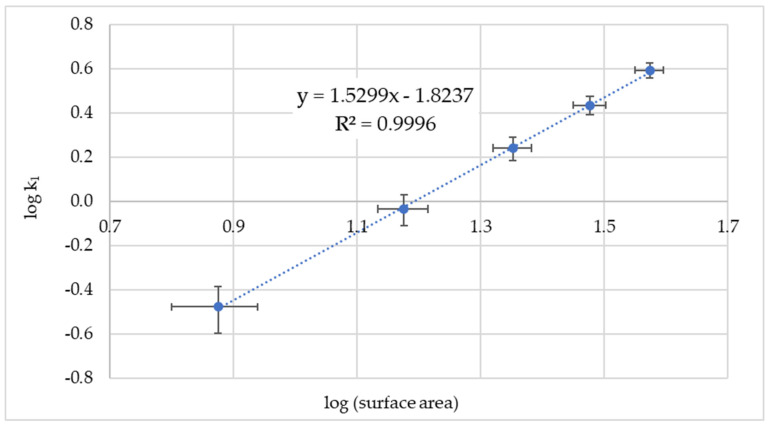
Plot of logk_1_ of Mn(VII) disappearance versus the logarithm of GNP surface area. The error bars are the mean standard deviations.

**Figure 5 nanomaterials-14-00140-f005:**
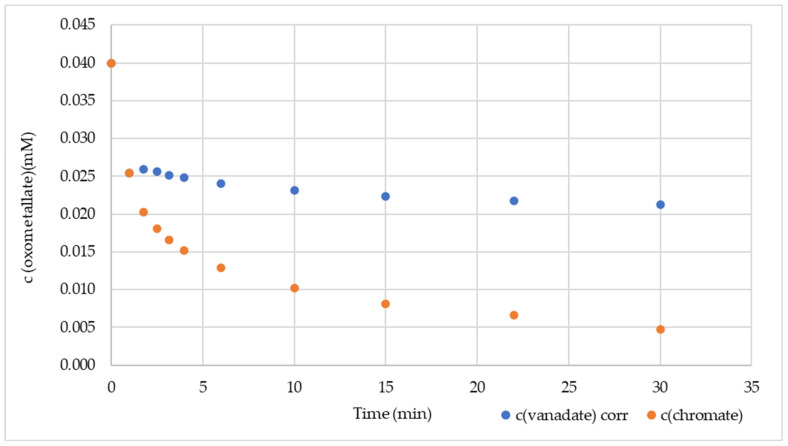
Concurrence between V(V) and Cr(VI) for 500 mg GNP in DW.

**Figure 6 nanomaterials-14-00140-f006:**
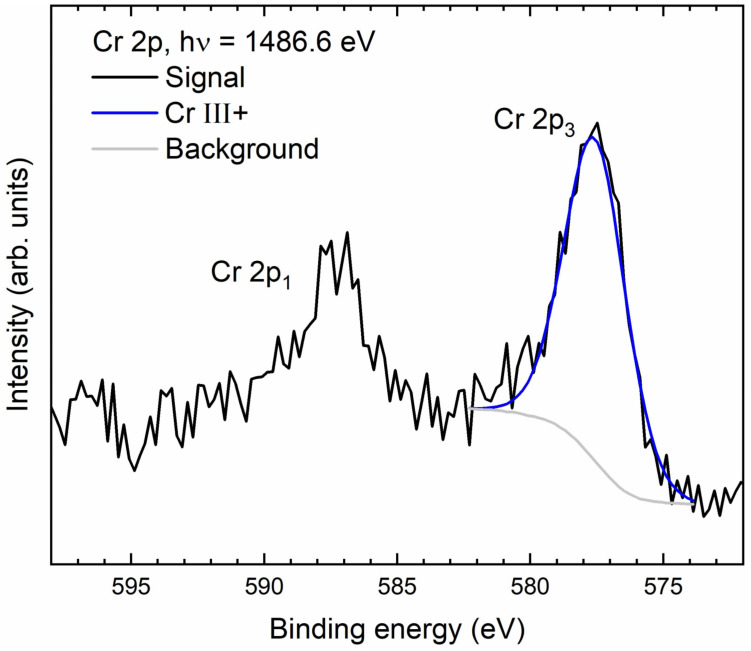
XP spectrum of chromium (Cr 2p) on GNP.

**Figure 7 nanomaterials-14-00140-f007:**
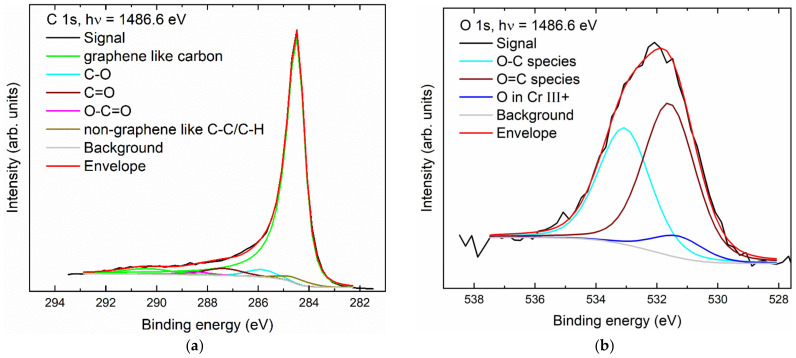
XP spectra of Cr-GNP: (**a**) carbon (C 1s); (**b**) oxygen (O 1s).

**Figure 8 nanomaterials-14-00140-f008:**
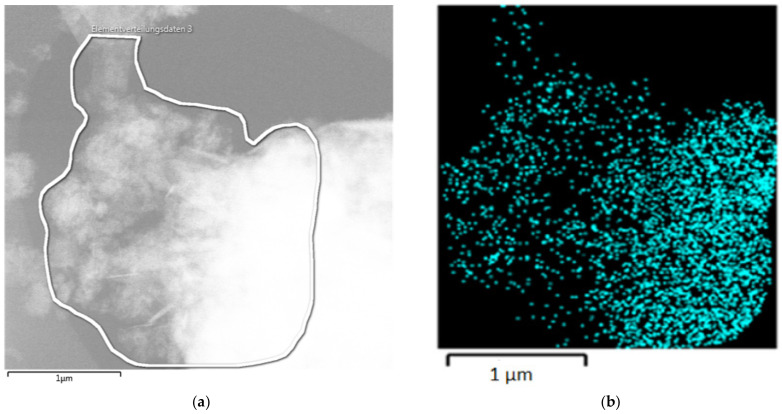
(**a**) TEM of a GNP flake; (**b**) the corresponding TEM-EDS for chromium (Cr Kα1).

**Figure 9 nanomaterials-14-00140-f009:**
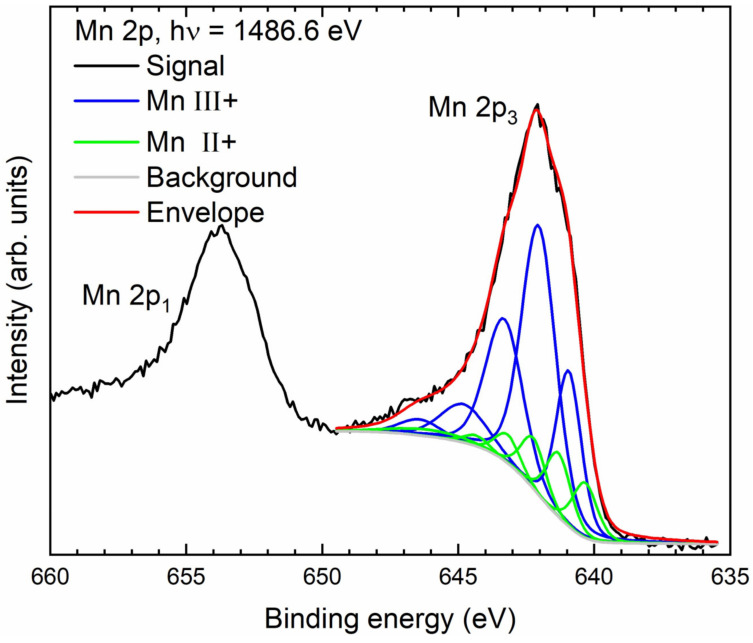
XP 2p signal of manganese on GNP with fit using the MnO and Mn_2_O_3_ line shapes according to [79].

**Figure 10 nanomaterials-14-00140-f010:**
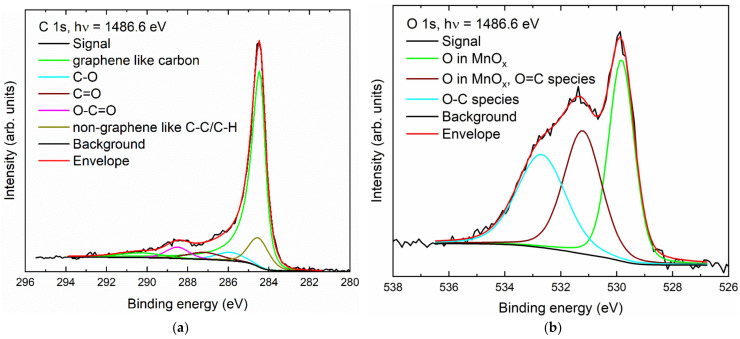
XP spectra of Mn-GNP: (**a**) carbon (C 1s); (**b**) oxygen (O 1s).

**Figure 11 nanomaterials-14-00140-f011:**
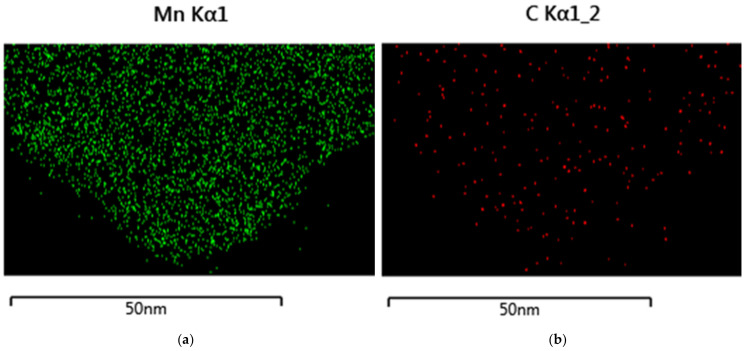
TEM-EDS of Mn-GNP. (**a**) TEM-EDS of manganese; (**b**) TEM-EDS of carbon.

**Figure 12 nanomaterials-14-00140-f012:**
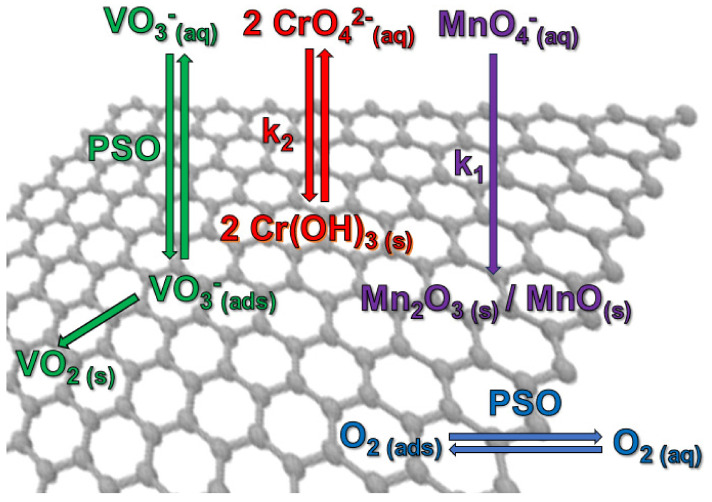
Visualization of the sorption processes of oxometalates and oxygen on GNP.

**Table 1 nanomaterials-14-00140-t001:** Charge transfer bands used for oxometalate quantification.

Oxometalate	Wavelength (nm)	Absorption Coefficient (L/mol × cm)	Literature
Vanadium (V) ^1^	263	1813	[58]
Chromium (VI) ^2^	375	4476	[59]
Manganese (VII)	525	2159	[60]

^1^ 266 nm [58], ^2^ 373 nm [59].

**Table 2 nanomaterials-14-00140-t002:** DLS measurements and calculations of GNP in DI and DW from 0.017 to 2000 μm.

Distribution Statistics	DI	DI (US) ^1^	DW	DW (US) ^1^
Mean (µm)	0.370	0.013	24.13	17.39
Median (µm)	0.291	0.098	25.75	20.41
Mode (µm)	0.271	0.073	31.50	21.70
Standard deviation (µm)	0.387	1.857	2.898	3.498
Cumulative results				
Number < 10%	0.169	0.059	5.815	3.202
Number < 25%	0.215	0.073	12.05	8.657
Number < 50%	0.291	0.098	25.75	20.41
Number < 75%	0.415	0.157	46.46	44.25
Number < 90%	0.609	0.269	76.84	79.76

^1^ US: with ultrasound treatment.

**Table 3 nanomaterials-14-00140-t003:** Effect of GNP and GNP reg on water pH and electric conductivity.

GNP in Water Dispersion	∆ pH	∆ E.C. (µS/cm)
50 mg GNP in DW	−0.11	−4
500 mg GNP in DW	−0.74	−34
50 mg GNP in SW (3.1 mM Na_2_SO_4_)	−1.71	1
500 mg GNP in SW (5 mM NH_4_Cl)	−2.67	12
50 mg GNP reg in DW	0.11	−1
500 mg GNP reg in DW	0.56	1
500 mg GNP reg in SW (5 mM NH_4_Cl)	0.47	−7
50 mg GNP reg in SW (3.1 mM Na_2_SO_4_)	0.86	3

**Table 4 nanomaterials-14-00140-t004:** PSO kinetic assessment of oxygen sorption on GNP at (24.4 ± 0.5) °C.

Process	Evaluation Time (min)	k_PSO_ (mM^−1^ × min^−1^)	q_e_ (mmol × L^−1^), Calculated	R^2^ Adj	pH Start	∆pH	∆E.C. (µS/cm)	∆O_2_ (mg/L)
0.25 mM O_2_ in 150 mL DW with 300 mg GNP	4.3–20.0	0.50 ± 0.10	0.070 ± 0.005	0.9571	7.03	−0.10	−3	−0.60
0.25 mM O_2_ in 150 mL DW with 300 mg GNP reg	6.0–13.0	2.09 ± 0.16	0.152 ± 0.002	0.9442	8.18	0.00	−1	−0.74
0.25 mM O_2_ in 150 mL SW with 300 mg GNP ^1^	9.2–19.0	0.29 ± 0.10	0.079 ± 0.010	0.9151	4.58	0.07	0	−0.32
0.25 mM O_2_ in 150 mL SW with 300 mg GNP reg ^1^	9.2–17.0	1.96 ± 0.11	0.135 ± 0.001	0.9529	7.60	0.00	0	−0.48

^1^ SW: 5 mM NH_4_Cl in DI.

**Table 5 nanomaterials-14-00140-t005:** Basic data for kinetic measurements of 0.04 mM oxometalates with GNP in drinking (DW) and synthetic water (SW).

Process	T_m_ (°C)	pH Start	ΔpH ^1^	Δ E.C. (µS/cm) ^1^	Exp. τ_1/2_ (min)	Capacity (mg/g)
0.04 mM NH_4_VO_3_ in DW, 500 mg GNP	24.8 ± 1.1	7.56	−0.54	−29	-	0.42
0.04 mM NH_4_VO_3_ in DW, 500 mg GNP reg	24.1 ± 0.1	7.63	0.53	1	4.6	0.75
0.04 mM NH_4_VO_3_ in 5 mM NH_4_Cl, 500 mg GNP	25.1 ± 0.1	7.34	−2.23	7	0.92	0.79
0.04 mM NH_4_VO_3_ in 5 mM NH_4_Cl, 500 mg GNP reg	25.2 ± 0.1	7.22	0.39	−10	1.7	0.87
0.04 mM K_2_CrO_4_ in DW, 500 mg GNP	24.8 ± 0.1	7.55	−0.50	−34	1.3	0.93
0.04 mM K_2_CrO_4_ in DW, 500 mg GNP reg	23.8 ± 0.1	7.48	0.61	2	0.80	1.0 ^2^
0.04 mM K_2_CrO_4_ in 5 mM NH_4_Cl, 507 mg GNP	25.2 ± 0.1	7.31	−1.71	−2	0.75	1.0
0.04 mM K_2_CrO_4_ in 5 mM NH_4_Cl, 500 mg GNP reg	24.5 ± 0.5	7.40	0.55	−21	0.60	1.0 ^3^
0.04 mM KMnO_4_ in DW, 50 mg GNP	24.1 ± 0.1	7.55	0.05	−5	0.33	10 ^4^
0.04 mM KMnO_4_ in DW, 50 mg GNP reg	24.0 ± 0.1	7.58	0.22	−2	0.45	11 ^4^
0.04 mM KMnO_4_ in 3.1 mM Na_2_SO_4_, 50.7 mg GNP	25.0 ± 0.1	7.26	−0.03	0	0.35	11
0.04 mM KMnO_4_ in 3.1 mM Na_2_SO_4_, 50.9 mg GNP reg	24.3 ± 0.3	6.85	2.21	1	0.33	11

^1^ Difference between end and start value, after initial time; ^2^ max. capacity 2.1 mg/g; ^3^ max. capacity 2.0 mg/g; ^4^ max. 300 mg/g.

**Table 6 nanomaterials-14-00140-t006:** Kinetic schemes used for data analysis.

Kinetic Type	Linearized Model Form	Linear Plot	Parameter ^1^	Equation
Pseudo first-order adsorption	$\ln\left[\frac{qe}{qe-q\left(t\right)}\right]=kt$	$\ln\left[\frac{qe}{qe-q\left(t\right)}\right]\;\;vs\;t$	q_e_: experimentally determined	(1)
			k_1_ = m	
Pseudo second-order adsorption	$\frac{1}{q\left(t\right)}=\frac{1}{k_{2}qe^{2}\left(\frac{1}{t}\right)}$	$\frac{1}{q\left(t\right)}\;vs\;\left(\frac{1}{t}\right)$	q_e_ = b^−1^	(2) ^2^
			k_2_ = b^2^/m	
Pseudo first-order reaction	$\ln c=-k_{1}t$	ln⁡c vs t	k_1_ = −m	(3)
Pseudo second-order reaction	$\frac{1}{c}=k_{2}t\;$	$\frac{1}{c}\;vs\;t\;$	k_2_ = m	(4)

^1^ m: slope; b: intercept; ^2^ [73].

**Table 7 nanomaterials-14-00140-t007:** Best fit kinetic data of 0.04 mM oxometalates with GNP in DW and SW.

Process	Kinetic Type	Data for Regression	Evaluation Time (Min) ^1^	k_PSO_ (mM^−1^ min^−1^) ^2^	q_e_ (mmol/L)	R^2^ Adj
0.04 mM NH_4_VO_3_ in DW, 500 mg GNP	PSO	31	5.8–16.5	22.4 ± 0.5	0.0179 ± 0.0001	0.9651
0.04 mM NH_4_VO_3_ in DW, 500 mg GNP reg	PSO	28	8.5–17.5	4.05 ± 0.15	0.0367 ± 0.0004	0.9524
0.04 mM NH_4_VO_3_ in 5 mM NH_4_Cl, 500 mg GNP	PSO	28	9.1–12.8	30.7 ± 0.8	0.0328 ± 0.0001	0.9119
0.04 mM NH_4_VO_3_ in 5 mM NH_4_Cl, 500 mg GNP reg	PSO	26	7.4–11.5	4.95 ± 0.22	0.0421 ± 0.0005	0.9367
				k_2_ (mM^−1^ min^−1^)		
0.04 mM K_2_CrO_4_ in DW, 500 mg GNP	Pseudo 2nd	31	8.0–23.0	6.91 ± 0.03	-	0.9970
0.04 mM K_2_CrO_4_ in DW, 500 mg GNP reg	Pseudo 2nd	29	9.4–18.0	446 ± 4	-	0.9831
0.04 mM K_2_CrO_4_ in 5 mM NH_4_Cl, 507 mg GNP	Pseudo 2nd	19	4.9–5.8	108 ± 4	-	0.9059
0.04 mM K_2_CrO_4_ in 5 mM NH_4_Cl, 500 mg GNP reg	Pseudo 2nd	19	5.1–5.7	484 ± 9	-	0.9762
				k_1_ (min^−1^)		
0.04 mM KMnO_4_ in DW, 50.0 mg GNP	Pseudo 1st	34	3.8–4.9	3.50 ± 0.01	-	0.9966
0.04 mM KMnO_4_ in DW, 50.0 mg GNP reg	Pseudo 1st	29	4.1–5.0	3.63 ± 0.02	-	0.9956
0.04 mM KMnO_4_ in 3.1 mM Na_2_SO_4_, 50.7 mg GNP	Pseudo 1st	31	3.8–4.8	2.23 ± 0.05	-	0.9022
0.04 mM KMnO_4_ in 3.1 mM Na_2_SO_4_, 50.9 mg GNP reg	Pseudo 1st	35	3.7–4.4	3.25 ± 0.04	-	0.9593

^1^ Dosage in the third minute. ^2^ Errors of rate constants and equilibrium concentrations (q_e_) were multiplied by the statistical factor of 3.

**Table 8 nanomaterials-14-00140-t008:** Elemental analysis of Cr-GNP with XPS.

Cr-GNP	C (%)	O (%)	Cr (%)	Ca (%)
Atom%	88.6	9.5	0.7	0.5
Mass%	83.6	11.9	2.9	1.6

**Table 9 nanomaterials-14-00140-t009:** Elemental analysis of Mn-GNP with XPS.

Mn-GNP	C (%)	O (%)	Mn (%)	Ca (%)	Cl (%)
Atom%	60	27.0	10	0.7	2.5
Mass%	39.6	23.8	30.2	1.5	4.9

**Table 10 nanomaterials-14-00140-t010:** ORP of 0.04 mM oxometalates in drinking water (DW).

Oxometalate (0.04 mM in DW)	Mean ORP before Dosage (mV) ^1^	Mean ORP after Dosage (mV) ^1^	ORP Oxometalate (mV) ^2^
NH_4_VO_3_	266.7 ± 0.6	311.4 ± 4.0	44.6 ± 4.6
K_2_CrO_4_	242.1 ± 1.6	310.6 ± 1.0	68.5 ± 2.6
KMnO_4_	251.5 ± 1.8	758.0 ± 0.6	506.5 ± 2.3

^1^ Mean of 5 measurements with standard deviation multiplied by statistical factor 3; ^2^ differences between ORP after and before dosage.

**Table 11 nanomaterials-14-00140-t011:** Ratios of the rate constants of the oxometalate and the oxygen sorption on GNP/GNP reg.

GNP/Water Combination	k_PSO_ (V(V))/k_PSO_ (O_2_)	k_2_ (Cr(VI))/k_PSO_ (O_2_)
GNP in DW	44	13.8
GNP reg in DW	1.9	213
GNP in SW	106	372
GNP reg in SW	2.53	247

## Data Availability

Data are contained within the article and Appendix A.

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
