# Peer review of "Kinetics of Direct Reaction of Vanadate, Chromate, and Permanganate with Graphene Nanoplatelets for Use in Water Purification"

_nanomaterials, 2024, doi:10.3390/nano14020140_

Round 1
Reviewer 1 Report
Comments and Suggestions for Authors
This manuscript reported the adsorption of vanadium(V), chromium(VI) and manganese(VII) on graphene nanoplatelets (GNP) and analyzed the adsorption kinetics. The topic is interesting and the work is sufficient. However, there are some issues should be addressed before its acceptance.
1) In the abstract, the abbr (PSO) should be given its full name as it first appeared.
2) The quality of Figures 1-4 should be improved.
3) Fig.7, a is empty, and XP ?
4) The summary and outlook is too long.
Comments on the Quality of English LanguageThe English should be further polished.
Author Response
Dear Reviewer,
Thank you very much for your effort. We hope our answers will satisfy you.
1) In the abstract, the abbr (PSO) should be given its full name as it first appeared.
The recommendation was implemented and the abstract has been revised.
2) The quality of Figures 1-4 should be improved.
The quality of Figures 1-4 has been improved.
3) Fig.7, a is empty, and XP ?
Sorry, we replaced the pictures into tiff-files.
4) The summary and outlook is too long.
The paragraph has been shortened to a small extent. We believe that further shortage will not improve the item.
Comments on the Quality of English Language
The English should be further polished. The language and typography were checked and improved if necessary.
Kind regards and best wishes for the new year.
Norbert Konradt
Reviewer 2 Report
Comments and Suggestions for Authors
Author Response
Dear reviewer,
Many thanks for your effort. We hope our answers will satisfy you.
More recent references have been added.
The abstract has been revised.
Kind regards and best wishes for the new year.
Reviewer 3 Report
Comments and Suggestions for Authors
This important manuscript deserves better graphical representation. Authors should pay more attention to fitting of XPS. Why do you fit only one peak from spin-orbital splitting?
The manuscript is full of misprints, that should be corrected.
For example “Within 19 minutes, the GNP were covered with ~0.4 mg/g vanadium (chemical formula?) and ~1.0 mg/g chromium as Cr(OH)3.”
Comments on the Quality of English LanguageThe manuscript should be corrected befor publishing. There are typos in the text.
Author Response
Dear reviewer,
Many thanks for your effort. We hope our answers will satisfy you.
This important manuscript deserves better graphical representation. Authors should pay more attention to fitting of XPS. Why do you fit only one peak from spin-orbital splitting?
We thank the reviewer for the comments.
This is done in this publication in accordance with literature and databases on photoelectron spectropscopy. Some of these examples are by ThermoFisher -https://www.thermofisher.com/de/de/home/materials-science/learning-center/periodic-table/transition-metal/iron.html), by colleagues from the Western University of Canada (www.xpsfitting.com). A multitude of publications on the topic also show that this is a valid approach, some of which are also cited in our draft [78,79] but also others, for example for manganese Nesbitt and Banerjee, American Mineralogist, vol. 83, no. 3-4, 1998, pp. 305-315. https://doi.org/10.2138/am-1998-3-414, for chromium Bandara, P.C., Peña-Bahamonde, J. & Rodrigues, D.F. Redox mechanisms of conversion of Cr(VI) to Cr(III) by graphene oxide-polymer composite. Sci Rep 10, 9237 (2020). https://doi.org/10.1038/s41598-020-65534-8). We hope this clarifies and answers sufficiently the concerns regarding the fitting procedure.
The manuscript is full of misprints, that should be corrected. The misprints
For example “Within 19 minutes, the GNP were covered with ~0.4 mg/g vanadium (chemical formula?) and ~1.0 mg/g chromium as Cr(OH)3.”
While the compounds for the GNPs with chromium and manganese could be identified using XPS as chromium(III) hydroxide and manganese(II) and (III) oxide, this was not the case for vanadium. The decrease in vanadate in the suspension with GNP was detected photometrically and using ICP. The elemental analysis also showed that vanadium was sorbed by the GNP.
Comments on the Quality of English Language
The manuscript should be corrected befor publishing. There are typos in the text. The typography was checked and, if problematic, improved.
Warm greetings and best wishes for the new year.
Norbert Konradt